# An Emerging Paradigm for ABCC5/MRP5 Function in Human Physiology

**DOI:** 10.3390/ijms26189211

**Published:** 2025-09-20

**Authors:** Jenai Chinoy, Charlotte Meller, Heidi de Wet

**Affiliations:** 1Bedford Hospital South Wing, Kempston Road, Bedford MK42 9DJ, UK; jenaichinoy@doctors.org.uk; 2Department of Physiology, Anatomy and Genetics, University of Oxford, Oxford OX1 3PT, UK; charlotte.l.meller@diamond.ac.uk; 3Electron Bio-Imaging Centre (eBIC), Diamond Light Source, Harwell Science & Innovation Campus, Didcot OX11 0GD, UK; 4Research Complex at Harwell, Harwell Science & Innovation Campus, Didcot OX11 0GD, UK

**Keywords:** ABCC5, cMOAT, MRP5, ABC-transporters, NAAG, *Abcc5*^−/−^ mice, cAMP, cGMP, haem

## Abstract

Since the first paper published by Susan Cole in 1990 detailing multidrug resistance mediated by ABCC1/MRP1, research into the C-subfamily of ATP-binding cassette transporters has continued to uncover a wide range of functionally divergent proteins. However, several orphan transporters remain in the C-subfamily, and the physiological function and substrates of ABCC5, ABCC11, and ABCC12 remain elusive. This review explores the emerging understanding of human ABCC5. Unlike other ABC transporters with well-defined drug export functions, ABCC5’s physiological roles remain only partially understood. While it is known for its involvement in multidrug resistance in cancers, recent studies suggest broader implications in development, metabolism, neurobiology, and male fertility. ABCC5 exports various endogenous substrates, including cyclic nucleotides (cAMP and cGMP), glutamate conjugates like NAAG, and possibly haem. Knockout models in mice, zebrafish, and sea urchins reveal ABCC5’s role in gut formation, brain function, eye development, and iron metabolism. In mice, its deletion results in lower adipose tissue mass, enhanced insulin sensitivity, and neurobehavioral changes resembling schizophrenia, highlighting its role in glutamatergic signalling and circadian regulation. Functionally, ABCC5 appears to impact adipocyte differentiation and GLP-1 release, implicating it in type 2 diabetes susceptibility in humans. Structural studies using human ABCC5 revealed a novel autoinhibitory mechanism involving a peptide segment (C46–S64) that blocks substrate binding, offering new potential for selective inhibitor development. However, this review emphasises caution in targeting ABCC5 for cancer therapy due to its underappreciated physiological function(s), particularly in the brain and male reproductive system. Understanding ABCC5’s substrate specificity, regulatory mechanisms, and functional redundancy with its paralog ABCC12 remains critical for successful therapeutic strategies in humans.

## 1. Introduction

ABC transporters are a family of integral membrane ATPases that transport a large number of structurally unrelated compounds in and out of cells. Broadly, there are seven subfamilies of ABC transporters (ABCA–ABCG) found in all taxa of life, but the function and substrates of many ABC transporters remain elusive [1]. In mammals, the majority of ABC transporters act as exporters and generally serve the purpose of cellular bouncers, which carefully screen the small molecules allowed to enter the cells they guard [2]. ABC transporters, therefore, typically perform essential physiological barrier functions and are concentrated in the membranes of vulnerable tissues such as the testes, uterus, placenta, and blood–brain barrier, where they protect against environmental toxins, xenobiotics, and plant flavonoids [3]. Unsurprisingly, ABC transporters are most well known for their role in chemotherapy-resistant tumours; however, ABC transporters of the C-subfamily tend to deviate from the norm. For example, ABCC8 and -9 regulate inwardly rectifying potassium channels (Kir6.1 and Kir6.2) and form part of the K_ATP_ channel complex, while ABCC7, also known as CFTR, is a Cl^−^ channel [4,5]. ABCC4 does export a substrate, but the ABCC4 substrate prostaglandin E_2_ (PGE_2_) is a ligand for the G-protein-coupled receptor (GPCR) EP4, and ABCC4 activity, therefore, indirectly regulates ciliogenesis through a downstream GPCR protein kinase A (PKA)-coupled signal transduction cascade [6].

Although great progress has been made in elucidating the roles of many ABC transporters, several orphan transporters of unknown function remain, including ABCC5, also known as MRP5 or cMOAT. Human *ABCC5* gene expression is well established to be upregulated in several cancers, including breast, lung, liver, colorectal, pancreatic, ovarian, prostate, cervical, and nasopharyngeal tumours [7]. *ABCC5* gene overexpression in tumours is correlated with chemotherapy resistance and poor prognosis [7]. ABCC5 is in the same family as the known drug efflux pump ABCC1/MRP1 and is thought to confer chemoresistance through reducing effective concentrations of chemotherapeutic drugs in cancer cells. However, non-drug efflux mechanisms of chemoresistance have also been shown to operate in certain tumours overexpressing *ABCC5*, especially in the processes that drive prostate cancer metastases [8].

In humans, altered *ABCC5* gene expression levels are also linked to metabolism, obesity, type 2 diabetes, and adipocyte differentiation [9,10,11,12]. Furthermore, *ABCC5* has been linked via GWAS to cognitive ability, including memory and depression in humans, mice, and rats [13,14,15], and the ABCC5 protein is known to be expressed in human pyramidal neurons [16]. However, a mechanistic understanding of how ABCC5 activity at the protein level relates to these physiological functions remains entirely unknown. As excellent recent reviews thoroughly cover the multidrug resistance literature associated with ABCC5 [7], this review will focus specifically on the endogenous substrate(s) and physiological function(s) of ABCC5, as the exact role of ABCC5 in healthy mammalian physiology remains elusive.

In this review, we will aim to summarise the current knowledge of ABCC5 and its physiological role in mammalian energy homeostasis and brain health, with a specific focus on knockout models used to identify endogenous ligands and investigate the physiological function of Abcc5 in both vertebrates and invertebrates, as summarised in Figure 1. For clarity, the human gene and protein are referred to, by convention, as *ABCC5* and ABCC5, while *Abcc5* and Abcc5 are used throughout to indicate the Abcc5 gene and protein expression respectively in other model systems.

## 2. Endogenous Ligands

*ABCC5* is globally expressed in humans, but is specifically enriched in the brain, skeletal muscles, and heart [17]. The putative endogenous ligands for ABCC5 are shown in Table 1. Knockout animal models have further provided much-needed insight into the physiological roles of endogenous Abcc5 substrates.

### 2.1. Cyclic Nucleotides

Abcc5 has long been established as an exporter of cyclic nucleotides [18,19]. More recently, Abcc5-mediated cAMP export has been shown to play a central role in regulating gut invagination in sea urchin embryos [20], as well as adipocyte differentiation in mouse embryonic fibroblasts [11]. *Abcc5* expression peaks post-gastrulation in developing sea urchin embryos, and *Abcc5* knockdown embryos develop a hindgut prolapse in the later part of gastrulation. This phenotype can be rescued by cAMP supplementation, suggesting Abcc5 mediates hind gut invagination via a cAMP-dependent mechanism. It is important to note that direct cAMP transport by Abcc5 was not tested in this model system. In other words, cAMP transport by Abcc5 was inferred from the rescue phenotype observed after cAMP supplementation, and, therefore, does not rule out the possibility of coupled downstream GPCR signalling similar to that seen in Abcc4

Furthermore, Abcc5’s function in regulating development is well documented in vertebrates, with developing zebrafish embryos exhibiting significant time-dependent Abcc5 expression in brain, eye, liver, and intestinal tissues [21]. Indeed, overexpression of an Abcc5 dominant negative in developing zebrafish embryos leads to developmental retardation, with smaller heads, smaller eyes, overall reduction in body length, and pigmentation [22]. The Abcc5 substrate cGMP was shown to accumulate intracellularly in Abcc5-knockdown zebrafish embryos, where expression of the cell cycle arrest gene p21—known to be regulated by intracellular cGMP levels [23]—is significantly increased. These data support cyclic nucleotide-dependent Abcc5-mediated developmental regulation.

**Table 1 ijms-26-09211-t001:** Putative physiological substrates of ABCC5.

Substrate Type	Substrate Identified	2D Structure	References
Cyclic nucleotides	cAMP and cGMP Transport shown via vesicular transport assays with ^3^[H]-labelled substrates. Uptake quantified by scintillation counting of radio-labelled substrates.	cGMP 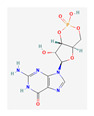 cAMP 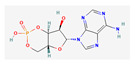	[18,19]
Nucleotide analogues	6-mercaptopurine, 6-thioguanine, and PMEA Transport shown via accumulation and efflux assays, and functional assays showing increased cellular resistance to cytotoxic drugs when overexpressing MRP5.	PMEA 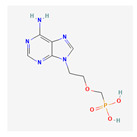	[24]
Glutamate conjugates/analogues	NAAG, NAAG_2_, BCG, BCG_2_ Asp-Gly-Glu, SAICAr, NAG, Val-Asp-Gly-Glu, and NAA NMDA, kainic acid, and domoic acid ZJ43 Transport shown via vesicular transport assays. Substrate uptake quantified by LC/MS.	NAAG 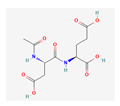	[25]
Metabolite	N-lactoyl amino acids Transport implied via functional assay showing reduced cellular growth of haem-dependent yeast cells overexpressing MRP5.	N-lactoyl-phenylalanine 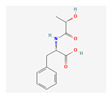	[26]
Iron homeostasis	Haem Transport implied via functional assay showing reduced cellular growth of haem-dependent yeast cells overexpressing MRP5.	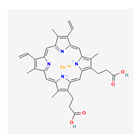	[27]
Folates	Folic acid Transport shown via vesicular transport assays with ^3^[H]-labelled substrates. Uptake quantified by scintillation counting of radio-labelled substrates.	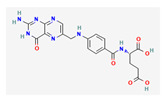	[19]
Antifolates	Methotrexate and methotrexate polyglutamine metabolites Transport shown via vesicular transport assays with ^3^[H]-labelled substrates. Uptake quantified by scintillation counting of radio-labelled substrates.	Methotrexate 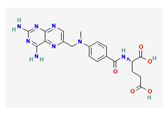	[19]

ABCC5 expression in human amniotic epithelia [28], cytotrophoblasts, and syncytiotrophoblasts [29,30] suggests a conserved role of ABCC5 in human development more broadly, although a specific role for ABCC5 in human development is still to be explored. A functional variant of Abcc5 in humans has been correlated with a decreased anterior chamber depth in the eye and a resultant increased risk of primary angle closure glaucoma [31]. This, alongside the smaller eyes seen in *Abcc5* KO zebrafish embryos, suggests a conserved role for Abcc5 in eye development and growth specifically. ABC transporters in general are highly expressed during embryogenesis [20], and given their function as active exporters in mammals, it is feasible to envisage a role for ABC transporters in establishing signalling molecule gradients and sinks essential for correct development.

### 2.2. Glutamate Conjugates

Jansen et al. (2015) identified several endogenous glutamate conjugates/analogues using untargeted metabolomics in murine *Abcc5* KO tissues [25]. This paper shows that Abcc5 activity regulates glutamate dipeptide availability in tissues, either directly by export or indirectly via regulation of synthesis/breakdown pathways. Notably, the glutamate dipeptide N-acetyl-aspartyl-glutamate (NAAG) was 2.4-fold higher in the brain of *Abcc5 KO* mice compared with wild-type mice. However, from these experiments, it was not possible to ascertain in which neurons or areas of the brain NAAG accumulated, and where NAAG and glutamate dipeptides accumulated in neurons. In other words, did NAAG accumulate inside neurons because Abcc5 is not present to load NAAG into synaptic vesicles, was it trapped in the synapse after release because it fails to be recycled, or was it building up in another cellular compartment or cell type?

In support of a possible neurological function in the brain, the de Wet laboratory demonstrated that mice genetically deficient for *Abcc5* are skinny and hyperactive, combined with altered memory consolidation and circadian rhythm disruption [32,33]. *Abcc5*^−/−^ mice exhibit a schizophrenia-like phenotype, with impaired sensorimotor gating (diminished pre-pulse inhibition), altered learning and memory processing (diminished fear-conditioning responses), and circadian rhythm disruption (delay in activity cycle peak, with reduced hours of sleep and less fragmented sleep). Further electrophysiological studies in *Abcc5*^−/−^ mice demonstrated altered N-methyl-D-aspartate receptor (NMDAR) current amplitudes and an increased spontaneous excitatory post-synaptic current frequency, all of which suggest that a loss of Abcc5 affects pre-synaptic glutaminergic signalling. But how Abcc5 regulation of glutamate neurotransmission relates to its function as an efflux pump remains unknown. As discussed above, the loss of Abcc5 impacts the availability of glutamate dipeptides in the brain, notably NAAG. NAAG is the third most prevalent neurotransmitter in the brain, and it inhibits glutamate neurotransmission via binding pre-synaptic mGluR3, which signals to downregulate pre-synaptic glutamate release [34]. NAAG has also been shown to modulate post-synaptic NMDAR signalling in a concentration-dependent way [35,36,37] and is, as such, neuroprotective. The role of specifically NAAG in schizophrenia remains controversial [38]. However, it has been previously proposed that antagonism of NMDAR receptors on recurrent, parvalbumin-positive GABAergic interneurons leads to pyramidal neuron disinhibition and subsequent glutamate excitotoxicity in schizophrenia [39]. Furthermore, a dual NMDAR hypofunction hypothesis has also been proposed, whereby this pyramidal neuron disinhibition in the early post-natal period leads to glutamate spillover, excitatory/inhibitory imbalances, and ensuing homeostatic pyramidal cell NMDAR downregulation, which ultimately may provide an explanation for the negative and cognitive symptoms of schizophrenia [40,41]. Interestingly, in line with the presentation of schizophrenia in humans, male *Abcc5*^−/−^ mice showed a much more pronounced cognitive phenotype compared with female littermates, and female *Abcc5*^−/−^ mice were able to compensate for the loss of Abcc5 activity to some degree [33]. However, it is important to note that neither NAAG nor NAAG_2_, nor the enzymes involved in their synthesis or breakdown, have been linked to schizophrenia in humans by recent GWAS studies [42].

Where and when NAAG accumulates in the brain, and the molecular mechanism by which glutamate di-peptide availability is regulated by Abcc5 activity, remains unknown. If and how Abcc5-dependent regulation of NAAG/glutamate dipeptide availability is linked to its effect on glutaminergic signalling and neurobiological phenotypes remains to be elucidated. However, it is clear that Abcc5 activity does affect glutaminergic signalling in mice and has clear neurological implications. As such, development of Abcc5 inhibitors should be approached with caution, as Abcc5 inhibitors could potentially have neurological side effects in mammals.

### 2.3. Haem

Haem has also been shown to be a possible endogenous ligand of Abcc5 in lower organisms [27], and Abcc5 activity is implicated in development and erythropoiesis in *Caenorhabditis elegans*. Haem is essential for *C. elegans*’s embryonic development as *C. elegans* is a haem auxotroph that does not perform de novo haem synthesis. Selective depletion of Abcc5 from the *C. elegans* intestine leads to embryonic lethality, a phenotype which can be rescued by dietary haem. The haem analogue ZnMP accumulates in the intestine of *Abcc5*-depleted *C. elegans*, collectively supporting an essential role for Abcc5 in exporting haem from the gut to extra-intestinal tissues, such as the embryo, to aid its development. Whilst non-essential in vertebrates, the failure of erythropoiesis and prevalence of morphological defects in *Abcc5*-knockdown zebrafish embryos further support a conserved role for regulation of iron and/or haem metabolism by Abcc5 [21,27]. Indeed, Abcc5 localises to both the basolateral membrane and organelles of the secretory pathway in mouse embryonic fibroblasts, collectively supporting a more specialist role for Abcc5 in exporting haem both extracellularly and between the cytosol and intracellular organelles in higher-order vertebrates [27]. Intriguingly, two of the glutamate di-peptide Abcc5 substrates identified by Jansen et al. (2015) were β-citryl-glutamate and β-citryl-glutamate-glutamate [25]. BCG has been shown to bind to iron and form strong complexes with both ferric (Fe^3+^) and ferrous (Fe^2+^) ions, and BCG is proposed to be an iron carrier in vivo [43,44]. It is, therefore, important to consider a possible indirect role for Abcc5 in the regulation of iron homeostasis and haem synthesis in vertebrates, where haem response elements in the *Abcc5* promoter may still be active to enhance Abcc5 expression in response to anaemia and increased iron requirements.

## 3. ABC Paralogs

Whilst the above experiments support a conserved role for Abcc5 in intracellular haem trafficking in higher-order vertebrates, this finding is not recapitulated in model mouse lines. *Abcc5 KO* mice are generally healthy and fertile [25,32], which is not uncommon in ABC-knockout mouse models, where loss of one ABC transporter is compensated for through the upregulation of another. Existing evidence would suggest that Abcc12 can compensate for the loss of Abcc5 in mice. Indeed, only *Abcc5*/*Abcc12* double-knockout (DKO) mice showed a pronounced phenotype, with marked male reproductive deficiency with reduced sperm quality, and significantly higher incidences of penile prolapse and urogenital tract clogging [45]. Abcc5 and Abcc12 both localise to spermatozoa mitochondrial-associated membranes (MAMs), across which they are proposed to transport several metabolites, including haem. *Abcc5*/*12 DKO* spermatozoa have aberrant, highly vacuolated mitochondria with reduced mitochondrial membrane potentials, and exhibit altered expression profiles of metabolites and genes that affect pathways related to mitochondrial function and retinoic acid metabolism. Upregulation of retinoic acid-responsive mitochondrial damage pathways E1F2a and mTOR in *Abcc5*/*12 DKO* spermatozoa altogether suggests Abcc5/12 work in concert to regulate intracellular transport of metabolites and prevent mitochondrial damage, possibly by a retinoic acid-dependent mechanism. Indeed, dysregulated retinoic acid signalling, and mitochondrial dysfunction more broadly, are both well-established causes of defective spermatogenesis, sperm function, and male infertility. No studies thus far have identified any hypomorphic Abcc5/12 SNPs as risk alleles for male infertility in humans, and perhaps this correlation warrants exploration in future genome-wide association studies. Important to note is the implications of therapeutic inhibition of Abcc5, as Abcc5 inhibitors could potentially have implications for male fecundity, notably in unidentified genetic populations with hypomorphic Abcc12 SNPs.

Furthermore, the presence of an additional Abcc5 paralog, ABCC11, in humans but not mice, limits the insight we have into the potential consequence of human ABCC5 loss of function from mouse studies alone.

## 4. Metabolism and Adipocyte Differentiation

Abcc5 has also been shown to be involved in regulating metabolism and adipocyte differentiation, but, to date, the specific substrate and mechanism(s) remain elusive. Over-expression of a human ABCC5 variant in subcutaneous fat was shown to correlate with T2DM (three-fold risk increase) and related phenotypes (beta cell function, peripheral insulin sensitivity, fasting IGR, and visceral fat) in European and African populations [9]. As would be expected from overexpression data, *Abcc5 KO* mouse models have a healthier metabolic phenotype, with reduced white and brown fat mass (in the absence of hypophagia) and increased insulin sensitivity [32]. The metabolic phenotype of global *Abcc5 KO* mice was, however, complicated to interpret, as incretin hormone release was also altered in these mice. Abcc5 expression was shown to be inversely correlated with GLP-1 release from L cells, which would explain the metabolically healthy *Abcc5 KO* phenotype. Gut hormones, and specifically incretin hormone (GIP and GLP-1) release, are regulated by multiple factors, such as diet, glutamatergic signalling, and gut pH, and the exact underlying mechanism of how the loss of Abcc5 activity regulates key hormones associated with metabolism remains unknown and may be complicated to dissect with global KO mouse models [46,47,48].

Further supporting a role of Abcc5 in adipocytes, a study using differentiated 3T3-L1 cells as an adipocyte model demonstrated that Abcc5 seems to function as a regulator of fat cell differentiation [11]. Both Abcc5 and Abcc4 are expressed during adipocyte differentiation, and a double knockdown of both *Abcc4* and *Abcc5* by siRNA promotes adipogenesis, but not synergistically. Interestingly, Abcc5 siRNA knockdown correlates with reduced *Abcc4* gene expression in adipocyte cells and vice versa, suggesting potential crosstalk between the two genes in regulating adipocyte differentiation. The study showed that *Abcc5* knockdown inhibits cellular efflux of its substrate cAMP, which was reported to promote upregulation of adipogenic genes PPARY/C/EBPalpha via both increased intracellular cAMP and Abcc4-mediated extracellular PGE availability, but these data would contradict the skinny phenotype observed in *Abcc5*^−/−^ mice. Furthermore, altered Abcc5 protein expression levels as a result of siRNA knockdown were not confirmed in this study. However, this study does highlight the complexity of the involvement of ABC transporters in adipogenesis, as contrary to the *Abcc5*^−/−^ mouse model, *Abcc4*^−/−^ mice exhibit a metabolically unhealthy phenotype, with impaired glucose tolerance, raised leptin levels, and adipocyte hypertrophy [49], but, to date, no double *Abcc5*^−/−^ and *Abcc4*^−/−^ mouse model exists. Taken together, these studies would suggest that Abcc5 is an important regulator of fat cell differentiation, and perhaps its overexpression confers a diabetes risk through disruptive lipid handling.

## 5. ABCC5 Structure

Prior to 2024, all mechanistic work on Abcc5 was undertaken in the absence of a high-resolution structure of the protein, and the interpretation of results was, therefore, somewhat limited. However, a recent ABCC5 structure was determined at a high resolution in both its inward-facing and ATP-bound outward-open conformations to 2.9 Å and 3.5 Å, respectively, enabling side-chain assignment. Interestingly, the structural determination of the inwards-facing state revealed the presence of a peptide bound in the central substrate-binding cavity (Figure 2A), which was assigned as the N-terminus residues C46-S64 of ABCC5. Consequently, an autoinhibitory mechanism by the transporter has been proposed whereby the substrate binding cavity of the transporter is initially blocked through binding of residues C46-S64, and only upon release of this can substrates bind and be exported by ABCC5, in a novel mechanism to ABC transporters. The binding of residues C46-S64 in the substrate-binding site is stabilised through several interactions, including three salt bridges.

The helix is also potentially stabilised through interaction of E51 and R55. Interestingly, there is conservation of the residues involved in these interactions across ABCC5 homologs (Figure 2B), supporting a key role of C46-S64 in the function of ABCC5. Crucially, the mechanism triggering the proposed release of C46-S64 is unknown, and the ATPase activities of wt-ABCC5 and hABCC5-Δ1-94 do not differ, which would suggest that occupancy of the substrate-binding site by the N-terminus C46-S64 peptide and ATPase hydrolysis are uncoupled [45]. More work is required to fully elucidate and validate the mechanisms of C46-S64 binding, the potential physiological implications of the proposed autoinhibition, and the impact on transport of ligands by ABCC5.

The authors further demonstrated that a designed peptide, M5PI, was capable of mimicking the binding of C46-S64, as demonstrated through further structural characterisation (Figure 2). Inhibition of ABCC5 transport activity was shown in *Xenopus laevis* oocytes, where M5PI expression, along with ABCC5, resulted in higher doxorubicin retention, a demonstrated substrate of ABCC5, in preloaded oocytes compared with ABCC5 expressed alone [45]. Consequently, this work, along with the structural determination, provides a potential peptide scaffold for the development of inhibitors of ABCC5. Interestingly, C46-S64 has a much larger volume than the proposed ligands of ABCC5 (Table 2), which would support the theory of autoinhibition by the transporter. Furthermore, haem is much larger than other proposed physiological substrates (Table 2), and it, therefore, cannot be ruled out that the proposed regulation of haem by ABCC5 is a secondary effect, with haem not directly transported by ABCC5.

Despite these molecular insights into ABCC5 transport gained through this study, direct structural observation of binding by a physiological ligand or the direct measurement of ligand transport by the transporter, such as purified protein in proteoliposomes, has yet to be demonstrated. Given the large size of the C46-S64 peptide in comparison with the proposed physiological substrates of ABCC5, observation of the binding of ligands would help identify key residues involved in binding to these, and comparison with the interactions of the C46-S64 peptide. These observations could aid in understanding the mechanism of C64-S64 binding and release, as well as explaining the promiscuity of transport by ABCC5. Interestingly, all the proposed substrates of ABCC5 contain two carbonyl groups in proximity to each other, perhaps suggesting these groups are involved in the specificity of substrate recognition by the transporter. Further structural insights into substrate binding by the transporter would be required to investigate this. Additionally, it would be interesting to observe the activity of purified ABCC5 reconstituted into proteoliposomes with different proposed radiolabelled ligands, to understand the ligand specificity of the system, and to investigate the suggested inhibition of ABCC5 by C46-S64.

Comparison of the sequence of homologs of human ABCC5 reveals that mouse and zebrafish homologs show 95% and 73.6% identity to human ABCC5 and show good sequence conservation in features of the protein key to substrate transport (Figure 2). However, *C. elegans* and sea urchin demonstrate 37.1% and 38.8% identity, respectively. Whilst this poor sequence conservation between homologs does not necessarily indicate differences in substrates of transport of Abcc5, there is, particularly, a lack of sequence conservation in the R-motif, demonstrated to be involved in cellular localisation of Abcc5 [50] (Figure 2C). Consequently, this should be considered when inferring human physiological function from cell-based studies based on *C. elegans* and sea urchin Abcc5.

## 6. Conclusions

Abcc5 is a highly conserved and ubiquitously expressed protein, with diverse exogenous and endogenous substrates. While it has a well-established role in conferring cancer chemotherapy resistance, it also has wide-ranging physiological functions in development, iron homeostasis, male reproductive function, adipose biology, and neurobiology, which remain poorly understood, and a mechanistic understanding of the physiological role of Abcc5 is still lacking (Figure 1). However, as the loss of Abcc5 in rodents leads to a clear neurobiological phenotype, and Abcc5 has an important role in male reproductive function in mice, it would be prudent to approach the use of ABCC5 inhibitors to treat multidrug-resistant prostate cancer in men with caution, as current data suggest that the inhibition of ABCC5 may lead to unexpected side effects.

## Figures and Tables

**Figure 1 ijms-26-09211-f001:**
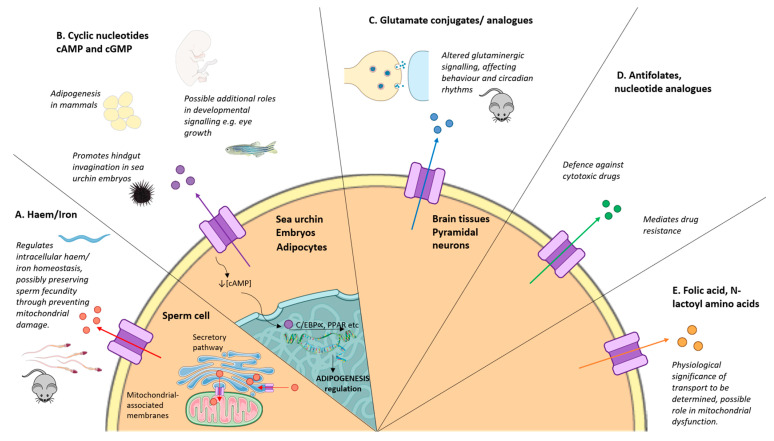
Physiological substrates and functions of Abcc5. (**A**) Abcc5 regulates haem availability in *C. elegans* and plays a role in haem and iron homeostasis in rodent sperm. (**B**) Abcc5 regulates cyclic nucleotide levels during developmental processes and has been implicated in adipogenesis in mammals, hindgut development in sea urchins, and eye development in humans and zebra fish. (**C**) Abcc5 is a putative glutamate dipeptide transporter, which impacts glutamatergic signalling, behaviour, and circadian rhythms in mammals. (**D**) Abcc5 overexpression is associated with drug resistance in numerous cancers. (**E**) Abcc5 transports N-lactoyl amino acids, which are associated with mitochondrial dysfunction, but the physiological implications remain largely unknown.

**Figure 2 ijms-26-09211-f002:**
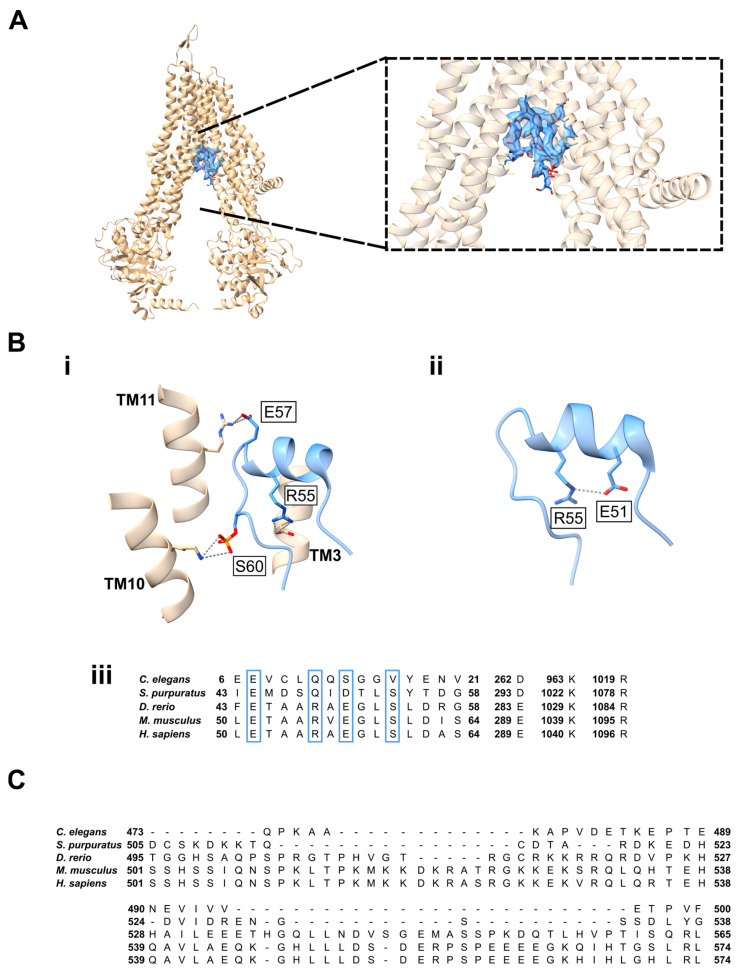
(**A**) The inwards-facing conformation of ABCC5 determined by Huang et al. (2025) ([50]; PDB:8WI0) showing the bound C46-S64 peptide. The map and model of the bound peptide are shown in blue. (**B**) (**i**,**ii**) Salt bridge interactions of the bound peptide. The sequence alignment of hABCC5 with homologs is shown (**iii**). (**C**) Sequence alignment of the R-motif of hABCC5 with homologs. Protein structure figures were prepared using [51].

**Table 2 ijms-26-09211-t002:** Approximate volumes of putative physiological substrates of ABCC5. The volumes were calculated using MoloVol 1.2.0 [52] using ligands found bound in experimentally determined protein structures.

Ligand	Volume (Å^3^)	PDB
C46-S64	1559	8WI0 [50]
Haem	661/512	5AZ3 [53]/1C52 [54]
NAAG	232/230	8U3G [55]/3BXM [56]
Folic acid	349/339	4LRH [57]/1DYI [58]
6-mercaptopurine	118/120	3BGD [59]/3NS1 [60]
cAMP	230/235	1HW5 [61]/1LPC [62]
Methotrexate	368/364	1AXW [63]/1DF7 [64]

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
