# Peer review of "An Emerging Paradigm for ABCC5/MRP5 Function in Human Physiology"

_ijms, 2025, doi:10.3390/ijms26189211_

Round 1

Reviewer 1 Report

Comments and Suggestions for Authors

Synopsis:
The subfamily C of ABC transporters is unique in having non-transporter functions and containing many orphan genes whose functions are still undefined. Chinoy et al wrote a review article aiming to provide an update on one of the orphan ABCC members, ABCC5. While its high-resolution models have become available recently, its role in cells and human physiology is still largely unknown. The manuscript was well written and straightforward to follow. The following are some suggestions for minor revisions before finalizing the publication. Hope it helps.

Suggestions:
1. There are some typos, likely unintentional, such as “MDR1” instead of MRP1.

2. Section 2 lists several putative endogenous transport ligands. It would help to stress whether any was ever confirmed by some transport assays. In addition, all these molecules seem to include at least two carbonyl groups in proximity. Is this something that the authors could elaborate?

3. Section 4 discusses the roles of ABCC5 and ABCC4 in adipocyte biology. Abcc5-/- and Abcc4-/- mouse models were reported independently, so unless there were data from a double KO model, it is not clear whether these two genes work together.

4. Section 5 starts the discussion about the autoinhibition by the N-terminal C46-S64 shown in the recent cryo-EM structures. Before the structural data was available, if there were other reports showing autoinhibition, it would help to cite those works to strengthen this section about the structural data.

5. At the end, the authors argued that the low sequence identity from worms or sea urchins would not support the ability to infer human physiology using Abcc5 studies from other organisms. Why not? Many functional homologs share almost identical protein structures despite low sequence identity. The fact that conserved residues are shown around the C46-S65 peptide binding suggests that Abcc5 from C elegans or sea urchin may share the same overall molecular structure. The difference may lie in the less conserved regions, such as the R-motif, whose cellular function could be species-dependent.

Author Response

Comments and Suggestions for Authors

Reviewer 1

The subfamily C of ABC transporters is unique in having non-transporter functions and containing many orphan genes whose functions are still undefined. Chinoy et al wrote a review article aiming to provide an update on one of the orphan ABCC members, ABCC5. While its high-resolution models have become available recently, its role in cells and human physiology is still largely unknown. The manuscript was well written and straightforward to follow. The following are some suggestions for minor revisions before finalizing the publication. Hope it helps.

We would like to thank the reviewer for their careful reading of the document and helpful feedback, which leaves the document much improved.  Changes suggested by Reviewer 1 is highlighted in the document in Yellow.

Suggestions:
1. There are some typos, likely unintentional, such as “MDR1” instead of MRP1.

We have now proofread the document very carefully and had made corrections throughout, highlighted in yellow in the text.  The document is currently written in UK English.

  1. Section 2 lists several putative endogenous transport ligands. It would help to stress whether any (i) was ever confirmed by some transport assays. (ii) In addition, all these molecules seem to include at least two carbonyl groups in proximity. Is this something that the authors could elaborate?

(i)  We have added information to Table 1 about the experimental methods used to verify the transport of each listed substrate of ABCC5. 

((ii)  We have also noticed the presence of the two carbonyl groups present in all proposed substrates, but without in-depth modelling any real conclusions can’t be drawn.  Unfortunately, no high-resolution structures exist of Abcc5 bound to a substrate.  But, in order to address the presence of carbonyl groups proposed endogenous ligands, we have modified the text to specifically mention this observation and to discuss future experimentation that is needed to elucidate this importance:

These observations could aid in understanding the mechanism of C64-S64 binding and release as well as explaining the promiscuity of transport by ABCC5.  Interestingly, all the proposed substrates of ABCC5 contain two carbonyl groups in proximity to each other, perhaps suggesting these groups are involved in specificity of substrate recognition by the transporter.  Further structural insights into substrate binding by the transporter would be required to investigate this.  Additionally, it would be interesting to observe the activity of purified ABCC5 reconstituted into proteoliposomes with different proposed radiolabelled ligands, to understand ligand specificity of the system, and to investigate suggested inhibition of ABCC5 by C46-S64.

  1. Section 4 discusses the roles of ABCC5 and ABCC4 in adipocyte biology. Abcc5-/- and Abcc4-/- mouse models were reported independently, so unless there were data from a double KO model, it is not clear whether these two genes work together.

We agree with the reviewer, there is no double Abcc4 and Abcc5 ko mouse model.  However, Laddha et al., do simultaneously knock down both C4 and C5 in 3T3-L1 cells using siRNA, and find there to be no synergistic affect on adipogenesis.  They also find that C4 knockdown correlates with reduced expression of C5 and vice versa, suggesting that both proteins may regulate each other’s’ expression, although direct evidence is not provided.  In order to clarify this section, and emphasise the fact that there are no double ko mouse models, we have now included the following sentences in the relevant section:

Further supporting a role of Abcc5 in adipocytes, a study using differentiated 3T3-L1 cells as an adipocyte model, demonstrate that Abcc5 seems to function as a regulator of fat cell differentiation (11).  Both Abcc5 and Abcc4 are expressed during adipocyte differentiation, and a double knockdown of both Abcc4 and Abcc5 by siRNA promote adipogenesis, but not synergistically.  The study showed that Abcc5 knockdown inhibits cellular efflux of its substrate cAMP, which was reported to promote upregulation of adipogenic genes PPARY/C/EBPalpha via both increased intracellular cAMP and Abcc4-mediated extracellular PGE availability, but this data would contradict the skinny phenotype observed in Abcc5-/- mice.  Furthermore, altered Abcc5 protein expression levels as a result of siRNA knock-down were not confirmed in this study.  However, this study does highlight the complexity of the involvement of ABC transporters in adipogenesis, as contrary to the Abcc5-/- mouse model, Abcc4-/- mice exhibits a metabolically unhealthy phenotype – with impaired glucose tolerance, raised leptin levels and adipocyte hypertrophy (49), but to date no double Abcc5-/-, Abcc4-/-.mouse model exists.  Taken together, these studies would suggest that Abcc5 is an important regulator of fat cell differentiation, and perhaps its overexpression confers a diabetes risk through disruptive lipid handling.

  1. Section 5 starts the discussion about the autoinhibition by the N-terminal C46-S64 shown in the recent cryo-EM structures. Before the structural data was available, if there were other reports showing autoinhibition, it would help to cite those works to strengthen this section about the structural data.

As far as we are aware, no previous reports of autoinhibition of ABCC5 were made prior to the structural observation of C46-S64 bound to the transporter.

  1. At the end, the authors argued that the low sequence identity from worms or sea urchins would not support the ability to infer human physiology using Abcc5 studies from other organisms. Why not? Many functional homologs share almost identical protein structures despite low sequence identity. The fact that conserved residues are shown around the C46-S65 peptide binding suggests that Abcc5 from C elegans or sea urchin may share the same overall molecular structure. The difference may lie in the less conserved regions, such as the R-motif, whose cellular function could be species-dependent.

We agree with the reviewer that the conserved residues of the binding pocket do suggest a similar molecular structure, and therefore substrates of transport across the homologs.  We have amended the text to focus this on discussion of the R motif of Abcc5 and its cellular localisation.

Comparison of the sequence of homologs of human ABCC5 reveals mouse and zebrafish homologs show 95% and 73.6% identity to human ABCC5 and show good sequence conservation in features of the protein key to substrate transport (Figure 1).  However, C. elegans and sea urchin demonstrate 37.1% and 38.8% identity, respectively. Whilst this poor sequence conservation between homologs does not necessarily indicate differences in substrates of transport of Abcc5, there is particularly lack of sequence conservation in the R motif, demonstrated to be involved in cellular localisation of Abcc5 (50) (Figure 1C). Consequently, this should be considered when inferring human physiological function from cell-based studies based of C. elegans and sea urchin Abcc5.

Reviewer 2 Report

Comments and Suggestions for Authors

The review by Chinoy et al is very precise and has some good survey of work on ABCC5 and could be useful for readers in the field. However, I have some suggestions that would make it better:

1: The review seems to be disjointed and the connection between different sections of manuscript is missing. There is lack of continuity between different sections.

2: Authors enlisted around 7 of the ligands of ABCC5 in table 1. However, they have detailed discussion about 3 of the enlisted substrate/ligand of ABCC5 in subsequent sections. The rational/logic of just discussing those 3 specific ligands in details are missing currently.

3: It will be useful if authors incorporate a pictorial cartoon of signaling events controlled/influenced by ABCC5 is cancer and other physiological events. Pictorial presentation of signaling/function by ABCC5 would be more helpful for readers to comprehend the role of ABCC5 under different physiological settings.

4: Authors might consider moving some sections of review like moving ABCC5 structure and functional relationship, just after introduction. 

Author Response

Reviewer 2

The review by Chinoy et al is very precise and has some good survey of work on ABCC5 and could be useful for readers in the field. However, I have some suggestions that would make it better:

We would like to thank the reviewer for their careful reading of the document and helpful feedback, which leaves the document much improved.  Changes suggested by Reviewer 2 is highlighted in the document in Blue.

1: The review seems to be disjointed and the connection between different sections of manuscript is missing. There is lack of continuity between different sections.

We have now rewritten the Abstract to better highlight the content and slant of the review.  We have also added further clarification in the Introduction text to better explain the format and approach taken by the review to orientate the reader; and have also added additional clarifying text to the end of each section to clearly signal the transition of ideas from one section to the next.

Abstract Since the first paper published by Susan Cole in 1990 detailing multi-drug resistance mediated by ABCC1/MRP1, research into the C-subfamily of ATP-binding cassette transporters continued to uncover a wide range of functionally divergent proteins.  However, several orphan transporters remain in the C-subfamily and the physiological function and substrates of ABCC5, ABCC11 and ABCC12 remains elusive.

This review explores the emerging understanding of human ABCC5.  Unlike other ABC transporters with well-defined drug-export functions, ABCC5’s physiological roles remain partially understood.  While it's known for its involvement in multidrug resistance in cancers, recent studies suggest broader implications in development, metabolism, neurobiology, and male fertility.

ABCC5 exports various endogenous substrates, including cyclic nucleotides (cAMP, cGMP), glutamate conjugates like NAAG, and possibly haem.  Knockout models in mice, zebrafish, and sea urchins reveal ABCC5’s role in gut formation, brain function, eye development, and iron metabolism.  In mice, its deletion results in leaner body mass, enhanced insulin sensitivity, and neurobehavioral changes resembling schizophrenia, highlighting its role in glutamatergic signaling and circadian regulation.

Functionally, ABCC5 appears to impact adipocyte differentiation and GLP-1 release, implicating it in type 2 diabetes susceptibility.  Structural studies revealed a novel autoinhibitory mechanism involving a peptide segment (C46–S64) that blocks substrate binding, offering new potential for selective inhibitor development.

The review emphasizes caution in targeting ABCC5 for cancer therapy due to its underappreciated physiological function(s), particularly in the brain and male reproductive system.  Understanding its substrate specificity, regulatory mechanisms, and functional redundancy between ABCC5 and ABCC12 remains critical for successful therapeutic strategies in humans.

‘’In this review we will aim to summarise the current knowledge of ABCC5 and its physiological role in mammalian energy homeostasis and brain health, with a specific focus on knock-out models used to identify endogenous ligands and investigate the physiological function of Abcc5 in both vertebrates and invertebrates.’’

‘’Knock-out animal models have further provided much needed insight into the physiological roles of endogenous Abcc5 substrates.’’

‘’Abcc5 has also been shown to be involved in regulating metabolism and adipocyte differentiation, but to date the specific substrate and mechanism(s) remains elusive.’’

‘’Prior to 2024, all mechanistic work on Abcc5 had been done in the absence of a high-resolution structure of the protein, and the interpretation of results were therefore somewhat limited.’’ 

2: Authors enlisted around 7 of the ligands of ABCC5 in table 1. However, they have detailed discussion about 3 of the enlisted substrate/ligand of ABCC5 in subsequent sections. The rational/logic of just discussing those 3 specific ligands in details are missing currently.

The rationale of choosing the 3 specific ligands were based on the availability of knock-out animal models (sea urchin, c.elegans, zebra fish and mice).  In other words we specifically discuss the ligands for which animal models were available.  We have now highlighted this in the paper by adding the following sentence:

‘’Knock-out animal models have further provided much needed insight into the physiological roles of endogenous Abcc5 substrates.’’

3: It will be useful if authors incorporate a pictorial cartoon of signaling events controlled/influenced by ABCC5 is cancer and other physiological events. Pictorial presentation of signaling/function by ABCC5 would be more helpful for readers to comprehend the role of ABCC5 under different physiological settings.

We have now added Figure 2 to the paper, which can be seen in the attached file.

Figure 2.  Physiological substrates and functions of Abcc5. A) Abcc5 regulates haem availability in C.elegans and plays a role in haem and iron homeostasis in rodent sperm. B) Abcc5 regulates cyclic nucleotide levels during developmental processes and has been implicated in adipogenesis in mammals, hindgut development in sea urchins and eye development in humans and zebra fish. C) Abcc5 is a putative glutamate dipeptide transporter, which impacts on glutamatergic signalling, behaviour and circadian rhythms in mammals. D) Abcc5 overexpression is associated with drug resistance in numerous cancers. E) Abcc5 transports N-lactoyl amino acids, which is associated with mitochondrial dysfunction, but the physiological implications remain largely unknown.

4: Authors might consider moving some sections of review like moving ABCC5 structure and functional relationship, just after introduction. 

We have made several changes to the document to clarify the flow of ideas and tie the review together more comprehensively, as is detailed in section 1.  If the reviewer would allow, the authors would like to keep the sections in the order of the original submission and keep the ABCC5 structure section for last as this flows well into the conclusion paragraph, and also summarises the most recent and exiting progress in the ABCC5 research field. . 

Round 2

Reviewer 2 Report

Comments and Suggestions for Authors

The manuscript is significantly revised and my suggestions have been incorporated.

I just have one minor suggestion that Figure 2 if possible should be linked (referenced) to introduction section rather than in conclusion section so that it orients readers early on about different role of ABCC5.

Author Response

We have now moved the summary figure to the Introduction section, please see attached.

Much thanks, this change was very helpful and looks much better.